# An Electric Wheelchair Manipulating System Using SSVEP-Based BCI System

**DOI:** 10.3390/bios12100772

**Published:** 2022-09-20

**Authors:** Wen Chen, Shih-Kang Chen, Yi-Hung Liu, Yu-Jen Chen, Chin-Sheng Chen

**Affiliations:** 1Graduate Institute of Automation Technology, National Taipei University of Technology, Taipei 10608, Taiwan; 2Department of Mechatronics Control, Industrial Technology Research Institute, Hsinchu 310401, Taiwan; 3Department of Mechanical Engineering, National Taiwan University of Science and Technology, Taipei 106335, Taiwan; 4Department of Radiation Oncology, MacKay Memorial Hospital, Taipei 10449, Taiwan

**Keywords:** brain–computer interface (BCI), steady-state visual evoked potential (SSVEP), augmented reality (AR), canonical correlation analysis (CCA), electric wheelchair, simultaneous localization and mapping (SLAM)

## Abstract

Most people with motor disabilities use a joystick to control an electric wheelchair. However, those who suffer from multiple sclerosis or amyotrophic lateral sclerosis may require other methods to control an electric wheelchair. This study implements an electroencephalography (EEG)-based brain–computer interface (BCI) system and a steady-state visual evoked potential (SSVEP) to manipulate an electric wheelchair. While operating the human–machine interface, three types of SSVEP scenarios involving a real-time virtual stimulus are displayed on a monitor or mixed reality (MR) goggles to produce the EEG signals. Canonical correlation analysis (CCA) is used to classify the EEG signals into the corresponding class of command and the information transfer rate (ITR) is used to determine the effect. The experimental results show that the proposed SSVEP stimulus generates the EEG signals because of the high classification accuracy of CCA. This is used to control an electric wheelchair along a specific path. Simultaneous localization and mapping (SLAM) is the mapping method that is available in the robotic operating software (ROS) platform that is used for the wheelchair system for this study.

## 1. Introduction

With the advancement of science and technology, electric wheelchairs are widely used to help disabled people move to the destination they want to go. However, a person who suffers from a neurodegenerative disease such as amyotrophic lateral sclerosis (ALS), multiple sclerosis (MS) [1] (pp. 204–219), and so on may not be able to use an electric wheelchair because they find it difficult to use a joystick as the control. Therefore, an autonomous electric wheelchair with a navigation system [2,3] can be operated by these individuals. An electrode cap acquires electroencephalography (EEG) signals from the human brain and brain–computer interface (BCI) systems translate the EEG signals into motion commands in real time [4,5].

BCI systems have been developed over several years, and can record many types of neural signals non-invasively or invasively: these include EEG, functional magnetic resonance imaging (fMRI), microelectrode arrays (MEAs), intracortical recording, electrocorticography (ECoG) and so forth [6] (pp. 814–826). EEG and fMRI belong to the non-invasive BCI systems, and intracortical recording and ECoG are the two invasive modalities mainly used in BCI research [7] (p. 6285). This research focused on the non-invasive EEG-based BCIs that have been realized using a variety of paradigms, including steady-state visual evoked potential (SSVEP), P300, rapid serial visual presentation (RSVP), movement-related potentials (MRPs), motor imagery [8,9,10], etc. These paradigms can be identified in two groups: endogenous and exogenous. The exogenous BCI paradigms need to use an external scene to stimulate the brain cortex, such as flashing stimuli or auditory beeps, to evoke discriminative patterns in the brain. SSVEP, P300 and RSVP all belong to the exogenous BCI paradigms. The endogenous BCI paradigms, such as motor imagery and MRPs, produce brainwave signatures spontaneously without external stimulation [11]. SSVEP is widely used in the speller system because it has a high ITR, good signal-to-noise ratio (SNR) and allows users to select multiple targets. The P300 waveforms are the brain patterns that are commonly used for EEG-based BCIs in traditional studies. These involve rare, task-relevant events and are often recorded at a latency of approximately 300 ms after stimuli are enacted [12]. However, the literature shows that SSVEP is more accurate than P300. Although the ITR for a hybrid BCI combining P300 and SSVEP is higher than the ITR for SSVEP, the total time that is required for the experiment on P300 is significantly more than that for SSVEP [13] (p. 101, 884). This study uses SSVEP to present brain patterns. SSVEP scenarios involve several white flickers flashing rapidly on a black background. Repetitive visual stimuli can elicit them at frequencies from 1 to 100 Hz [14] (pp. 346–353). The screen display is an interface that shows the SSVEP scenarios for the user during the experiment [15] (pp. 614–627). This study uses three types of SSVEP scenarios; one controls the motion of an electric wheelchair for five types of motion: forward, left, right, backward and ending for an emergency, and a second uses room information for a destination, such as a room number and name. The third scenario gathers information about a room and creates a map to allow users to specify a destination more intuitively. If navigation is automatic, it may be restricted due to environmental factors [16] (pp. 128–139). Systems can account for environmental factors but converting the environmental map can require much time, so this study uses automatic navigation and users can control the direction of a vehicle.

EEG signals are acquired using an electrode cap that is positioned on the user’s scalp. These are interpreted using analytical methods or learning models, such as canonical correlation analysis (CCA), multivariate synchronization index (MSI) [17], support vector machine (SVM) [18], power spectral density analysis, k-nearest neighbors (k-NN) and linear discriminant analysis (LDA) [19,20]. The CCA method without channel location selection and parameter optimization analyzes the relationship between two samples of frequency information from multiple-channel EEG signals. Studies show that the CCA analyzes SSVEP signals and detects their frequencies better than other conventional recognition techniques [21,22]. Therefore, this study passes EEG signals through a bandpass filter to reduce interference and classification uses CCA to recognize SSVEP events.

The ITR is measured in bits per minute and is used to determine the communication speed for a BCI [23] (p. 025015). This study calculates ITR and classification accuracy using CCA to verify the online performance of the BCI system. Finally, the classification results from the CCA are then translated into motion commands and communicated to a robotic arm or an electric wheelchair. As an application of this paper, the classification results are used to produce motion commands and navigational instructions in real time to convey an electric wheelchair to a destination that is defined by the user [24].

Mapping is the most elemental application for an electric wheelchair with automatic navigation. Before the automatic navigation system is activated, an environment map must be established using sensors such as a camera, sonar or laser scanner that the wheelchair system uses to localize. The most common mapping method is SLAM, which creates a map while it localizes the robot’s location [25]. In this case, GMapping is an algorithm for SLAM that is used for this study. The navigation system is configured in the ROS environment to ensure safety and convenience.

Inspired by these researches, this research implements an EEG-based BCI system and an SSVEP to manipulate an electric wheelchair. The purpose of this system is to make life easier for people with reduced mobility. By using CCA as a tool for analyzing EEG signals, they can be more accurately identified as corresponding categories. The main contributions of this paper are as follows:For EEG signal analysis, two training-free algorithms, CCA and MSI, are applied in this article. The experimental results show that the accuracy rate is higher than 80.9%, so the proposed stimulus can be analyzed by various algorithms.This article improves the display stimuli, using MR goggles as a display tool, with significantly increased accuracy and improved space usage on the electric wheelchair.

The remainder of this paper is organized as follows. Section 2 presents the architecture for this study. Section 3 describes the design of the BCI system, including the SSVEP scenario and the method of analysis. Section 4 demonstrates the proposed BCI-based electric wheelchair control system. Section 5 details the experiment and the results are presented in Section 6.

## 2. Architecture

The EEG signals were collected from subjects who wore an EEG electrode cap and gazed at the stimulus of the scenario on a monitor or used mixed reality goggles. The three scenarios used correspond to different functions of the wheelchair system. The first scenario describes the direction of the wheelchair: there are four arrow-shaped patterns covered by four flickers, representing four directions and one square image with the word end covered by the remaining flicker. The second scenario uses five rectangles with information about the destination room. These are covered by the five different frequencies of flickers. The third uses an environment map with five rooms and five different frequencies of flickers overlapping each room image.

Before the EEG data is acquired, the environment map must be constructed by the wheelchair system. Scenario 3, which presents the appearance of the map, is then shown on the screen or mixed reality goggles to allow users to navigate to the designated locations. The user chooses a destination and a CCA-based classifier classifies the EEG signals into frequency classes to generate five confidence scores. The confidence scores represent the probability of the corresponding frequency of the EEG signal. The highest score indicates that this EEG signal is most related to this frequency and the electric wheelchair is moved according to the corresponding commands translated using the result of CCA. In other words, if the confidence score that corresponds to one of the frequency categories is significantly higher than the other scores, the destination coordinates that correspond to this frequency category are transmitted to the electric wheelchair to allow it to autonomously navigate to a location that is specified by the user. The program ends when the device reaches a destination. If all of the confidence scores are lower than the threshold, Scenario 2 replaces Scenario 3.

Scenario 2 uses the same five flickers as Scenario 3 and the EEG signal that corresponds to each flicker also produces five confidence scores after processing by the CCA classifier. If all of the confidence scores are less than a threshold, TR, Scenario 2 is replaced by Scenario 1 to allow the user to control the wheelchair by looking in the direction of travel.

Five confidence scores are also generated for Scenario 1. Similarly, if all of these scores are less than a threshold, TD, Scenario 3 is used for navigation. If one of the confidence scores is significantly higher than the others, the wheelchair moves in the direction that corresponds to this frequency. If the frequency category with the highest confidence score corresponds to 13 Hz with the word “end” the program ends. It also means that the user moves the electric wheelchair to the designated position and does not need to move. All in all, Scenario 2 and Scenario 3 allow automatic navigation and Scenario 1 allows users to manipulate the wheelchair. The architecture of the online system for this study is shown in Figure 1.

## 3. Brain–Computer Interface System

### 3.1. SSVEP Scenario Configuration and Design

To acquire the EEG datasets from the human brain, three different SSVEP scenarios are viewed by the user. The five flickers have specific frequencies (7, 8, 9, 11 and 13 Hz) that overlap on the figures and are configured at the four corners and the middle of the black background. There are two command modes to move the electric wheelchair: automatic mode or using human interface devices, such as a joystick or a keyboard.

For this study, Scenario 1 uses five color images to replace the joystick. There are four different orientation arrows and an end choice. Scenario 2 presents the information for each room, which the user can choose to directly move to the room. Scenario 3 uses a map that is constructed by the guidance system for the electric wheelchair. The flickers are separated to avoid interference when participants watch one of the targets. Scenarios 2 and 3 allow the electric wheelchair to move automatically. The configurations for these three scenarios are shown in Figure 2.

### 3.2. Canonical Correlation Analysis for EEG Signals

Among many recognition methods, SVM and CCA differ in that CCA does not divide data into trial and test data. The CCA method implements correlation maximization between the multichannel EEG signals. However, the 60 Hz AC noise in the environment is contained in the datasets that are collected by the EEG electrode cap. A four-order bandpass infinite impulse response filter eliminates interference and retains frequencies from 3 to 40 Hz. The filtered datasets are analyzed using CCA, which identifies and measures the associations between two sets of variables [26]. The relationship between each EEG signal that is collected by stimulating the visual cortex of the brain and the frequencies for the five classes, which are 7, 8, 9, 11 and 13 Hz, are compared with their harmonic frequency to calculate the classification accuracy. There are five classification rates for each EEG signal and the result with the highest accuracy determines the class of the dataset.

Assume two matrixes $X\in R^{n\times p}$ and $Y\in R^{m\times p}$ and define its cross-covariance matrix as $\sum_{XY}=cov(X,Y)$ which is m-by-n matrix. Find the vector $a\in R^{n}$ and $b\in R^{m}$ by canonical correlation analysis to maximize the correlation ρ=corr(U,V) between random variables $U=a^{T}X$ and $V=b^{T}Y$. Therefore, the ρ can be derived as
(1)$$\begin{array}{c} \rho =corr(U,V)\\ =\frac{cov(U,V)}{\sigma_{U}\sigma_{V}}\\ =\frac{a^{T}\sum_{XY}b}{\sqrt{a^{T}\sum_{XX}a}\sqrt{b^{T}\sum_{YY}b}} \end{array}$$

Then solve a and b in order to obtain the maximum solution for c; the conditions for optimizing this problem are defined as Equation (2)
(2)$$\begin{array}{c} \mathrm{Maximize}\;a^{T}\sum_{XY}b\\ \mathrm{Subject}\;\mathrm{to}\;a^{T}\sum_{XX}a=1,b^{T}\sum_{YY}b=1 \end{array}$$

According to the Lagrange multiplier method, Equation (3) is obtained
(3)$$L=a^{T}\sum_{XY}b-\frac{\lambda }{2}(a^{T}\sum_{XX}a-1)\;-\frac{\theta }{2}(b^{T}\sum_{YY}b-1)$$

Take the partial differential calculation of L with a and b, respectively, and let the two Equations be 0, as shown in Equations (4) and (5)
(4)$$\frac{\partial L}{\partial a}=\sum_{XY}b-\lambda \left(\sum_{XX}a\right)=0$$
(5)$$\frac{\partial L}{\partial b}=\sum_{YX}a-\theta \left(\sum_{YY}b\right)=0$$

Multiply the left side of Equation (6) by $a^{T}$ and the left side of Equation (7) by $b^{T}$, and then arrange it according to the constraints of Equation (2) to obtain Equation (8)
(6)$$a^{T}\sum_{XY}b-\lambda \left(a^{T}\sum_{XX}a\right)=0$$
(7)$$b^{T}\sum_{YX}a-\theta \left(b^{T}\sum_{YY}b\right)=0$$
(8)$$\Rightarrow \lambda =\theta =a^{T}\sum_{XY}b$$

According to the conditions of Equations (2) and (8), it is known that the desired λ is the maximum value, and Equations (6) and (7) are simplified and sorted into Equations (9) and (10)
(9)$${\sum_{XX}}^{-1}\sum_{XY}b=\lambda a$$
(10)$${\sum_{YY}}^{-1}\sum_{YX}a=\lambda b$$

Put (10) into (9) to obtain (11); you can find the eigenvalue $\lambda^{2}$ and the eigenvector a, and put (9) into (10) to obtain the eigenvector b
(11)$${\sum_{XX}}^{-1}\sum_{XY}{\sum_{YY}}^{-1}\sum_{YX}a=\lambda^{2}a$$
(12)$${\sum_{YY}}^{-1}\sum_{YX}{\sum_{XX}}^{-1}\sum_{XY}b=\lambda^{2}b$$

When λ is the maximum value, a and b at this time are called canonical variates, and λ is the maximum correlation coefficient between U and V, which is shown in Equation (13)
(13)ρ=corr(U,V)=λ

### 3.3. Multivariate Synchronization Index

MSI is an algorithm that can be directly analyzed without the training that CCA needs. This measure is to estimate the synchronization between the actual mixed signals and the reference signals as a potential index for recognizing the stimulus frequency [17,27]. We use the same filtering method as CCA to process the EEG signals before using MSI for analysis.

Assume an N by M matrix $X\in R^{N\times M}$ representing the filtered EEG signal. The MSI must also create a sample signal from the stimulus frequencies used in an SSVEP-based BCI system, similarly to CCA, and we assume a 2Nh by M matrix Y to represent it, where N is the number of channels used in the experiment, M is the number of sampling points of the EEG signal and Nh is the resonant frequency multiplier taken in the experiment. To find the synchronization index between the two sets of signals, the calculation is derived as follows

First the correlation matrix between X and Y must be calculated
(14)$$c=\left[\begin{array}{cc} c11 & c12\\ c21 & c22 \end{array}\right]$$
where
(15)$$c11=\frac{1}{M}XX^{T}$$
(16)$$c22=\frac{1}{M}YY^{T}$$
(17)$$c12=c21^{T}=\frac{1}{M}XY^{T}$$

Because this correlation matrix includes a cross-correlation matrix and an autocorrelation matrix, where the autocorrelation matrix will affect the synchronization calculation, we convert the autocorrelation matrix into a linear form for calculation to eliminate the influence
(18)$$C11=c11^{\frac{-1}{2}}$$
(19)$$C22=c22^{\frac{-1}{2}}$$
(20)$$U=\left[\begin{array}{cc} C11 & 0\\ 0 & C22 \end{array}\right]$$
(21)$$R=UCU^{T}=\left[\begin{array}{cc} I_{N\times N} & C11c12C22\\ C22c21C11 & I_{Nh\times Nh} \end{array}\right]$$

Let the matrix R obtained from Equation (21) have the number of K eigenvalues, where K=N+Nh. Then, the calculation to normalize all the eigenvalues $\lambda_{1}\ldots \lambda_{K}$ is as follows
(22)$$E=\frac{\lambda_{i}}{\sum_{i=1}^{K}\lambda_{i}}$$

Finally, calculate the max of synchronization index S as shown in Equation (23)
(23)$$S=1+\frac{\sum_{i=1}^{K}E_{i}log\left(E_{i}\right)}{log\left(K\right)}$$

### 3.4. Information Transfer Rate of EEG Signals

A standard measure for communication systems is the bit rate, which is the amount of information that is communicated per unit. The bit rate depends on speed and accuracy [28] (pp. 94–109). In order to determine whether the data are converted into a stimulus to increase accuracy, this study calculates the ITR for each flicker for different frequencies is shown in Figure 3. The ITR is calculated as
(24)$$B(bits/trial)=log_{2}n+plog_{2}p+(1-p)log_{2}\frac{1-p}{n-1}$$
(25)$$Q(trials/min)=\frac{S}{t}\times \frac{60}{1}$$
(26)ITR(bits/min)=B(bits/trial)×Q(trials/min)
where n is the number of targets (for this study, there are five classes in each scenario), p is the average value for the classification accuracy for five classes in each scenario, S is the number of trails and t is the average time for one selection, which includes the stimulation time and rest time before the following stimulus appears: the unit is seconds. For the experiment, the stimulation time and the rest time before the following stimulus is three seconds per trial is shown in Figure 4.

## 4. Electric Wheelchair Control

This study uses a human–machine interaction that uses brain signals to manipulate an electric wheelchair without a joystick. Therefore, the stimulus that replaces the joystick is described in the previous section. When the CCA reconstructs the brain signals into a corresponding control command, an interpreter translates the command into machine language ROS scripts to control the electric wheelchair.

### 4.1. Hardware and Software

The electric wheelchair for this study is shown in Figure 5. It is controlled using a laptop from ASUS in Taiwan with an Intel Core i7-6700HQ CPU, 8GB RAM and an NVIDIA GeForce GTX 960M GPU with 512 Tensor Cores. To reach a specified destination avoiding obstacles, an Intel^®^ RealSense D435i RGB-D camera from Intel Corporation, Santa Clara, California, United States and a SICK TIM551 2D-LiDAR from SICK, Waldkirch, Germany are used to determine the geography of the environment, as shown in Figure 6. Furthermore, the operating system is installed and configured in an ROS environment on Ubuntu 18.04 from Canonical company in UK on the PC.

### 4.2. Navigation

The concept of the autonomous navigation system is based on the mapping process using SLAM, which includes visual SLAM (V-SLAM) and Li-DAR-SLAM. For this study, the GMapping algorithm depicts the environment on a map using 2D-LiDAR before the electric wheelchair is driven to a specified destination. The current location of the electric wheelchair must be determined on the map. The current coordinate point is sent to SLAM to generate the navigational information by subscribing and publishing. Finally, the control command, which is the classification results from the EEG signals, is transmitted to the navigation system to complete the corresponding task.

The picture depicting the direction in which the wheelchair moves in Scenario 1 has five flickers that correspond to end, forward, backward, left and right turn. When the CCA classifier recognizes the class of the EEG signal corresponding to the flicker, the flicker that the subject observes is identified. For example, in Scenario 1, the flicker with a frequency of 7 Hz corresponds to the backward direction. When the CCA classifier recognizes the EEG data, the wheelchair receives a command to move backward.

The automatic navigation system is initiated if the second mode is used. The flicker with a frequency of 7 Hz corresponds to room “801A” in Scenario 2. After that, the wheelchair receives the navigation instruction, which is the location coordinates of room “801A”, and the map of field experimentation is used to allow the electric wheelchair to move to room “801A”.

Mode 3 triggers the automatic navigation system that allows the wheelchair to move autonomously. If the user does not understand the site map or the relative location of the room, Mode 3 generates an intuitive scenario for users. The participant looks at the screen for Scenario 3, which shows the environment map. The flicker with a frequency of 7 Hz also corresponds to room “801A” in Scenario 3, and the acquired EEG data are transmitted to the CCA classifier to verify the frequency.

## 5. Experiments

### 5.1. Experimental Setup

To ensure the accuracy of CCA classification in real-time system for every user, an experiment analyzed the acquired EEG signals for twelve subjects. During the data-acquisition phase, each subject performed 20 trials using each scenario that was displayed on the monitor and MR goggles. The total length of each recorded data sequence was 3 s.

This study acquired EEG signals using an OpenBCI Cyton board with OpenBCI software from Brooklyn, New York, USA and collected the EEG signals from a 21-channel EEG cap that has a sampling rate of 250 Hz. The scenarios were displayed on a monitor and MR goggles. An ASUS XG279Q, which is a high-level stimulating monitor with a 144 Hz refresh rate, and a Microsoft HoloLens 2 with a 60 Hz refresh rate were used to create stimuli for the BCI experiment. A red square overlapping on the flicker was used to specify the picture at which the participant looks.

The subject focused on the marked object and followed the instructions that were displayed on the screen or MR goggles to collect the data for frequencies of 7, 8, 9, 11 and 13 Hz. During the experiment using a screen, the participants sat 30 cm away from the screen and observed the flickers. The configuration for the experiment using a screen is shown in Figure 7. For the experiment that used MR goggles, participants wore an electrode cap kit and then put on the MR goggles. The configuration for the experiment using MR goggles is shown in Figure 8. Twelve subjects, ten males and two females, participated in the experiment. Their ages were in the range 39 ± 17. Each subject read and signed an informed consent form that was approved by the Study Ethics Committee for Human Study Protections (21MMHIS241e). When the data were acquired, they were fed into a CCA classifier to determine the classification accuracy.

When the CCA classification rate was verified, the electric wheelchair was operated online. When controlling the electric wheelchair using the real-time system, the subject is absorbed in the interface or MR goggles and has five options that correspond to respective flickers. This EEG signal measures the associations with the classified dataset. The outcome is translated into a command to control the movement of the wheelchair or a target location and is transmitted via TCP/IP.

### 5.2. SSVEP Experimental Results

This study collected the EEG signals from a 21-channel EEG cap that has a sampling rate of 250 Hz. A laptop configured with an Intel i7-10750H CPU, 16GB RAM and RTX 1660Ti 6GB GPU was used to acquire the EEG signals from the amplifier. The configuration of the channels on the EEG cap is shown in Figure 9. The three channels O_1_, O_2_ and Pz, in the occipital region, which is the yellow region in Figure 9, were used as the CCA classifier’s input signal to reconstruct the SSVEP-based stimulus.

Three scenarios describe the direction (Scenario 1), room information (Scenario 2), and environmental map (Scenario 3), which were analyzed using the CCA and MSI classifier. One experiment used a screen to present the scenarios and the other used MR goggles to display the flickers. The results of the first experiment using a screen and the classification method CCA, collected from twelve participants, are shown in Table 1, Table 2 and Table 3, respectively. The results of another experiment using MSI as an analysis tool are shown in Table 4, Table 5 and Table 6, respectively.

Scenario 1 was designed as a similar function as a joystick, used to control the direction of an electric wheelchair and the accuracies (ACCs) of CCA for four orientations, backward, forward, left and right and an end option, at frequencies of 7 Hz, 8 Hz, 9 Hz, 11 Hz and 13 Hz were 95%, 90.8%, 90.4%, 94.2% and 83%, respectively. The average ACC for all frequencies was 90.7%.

For the experiment involving automatic control five pictures of the room tag with the rooms’ names and identification numbers were used. For Scenario 2, the same frequencies were used, and the classification ACCs of CCA were 94.6%, 92.1%, 92.1%, 88.3% and 77.1%, respectively. The average ACC for all frequencies was 88.8%.

Scenario 3 used a map of the entire experimental field so the subject selected the location of the rooms directly. The ACCs of CCA were 87.1%, 82.5%, 87.1%, 82.1% and 76.7%, respectively. The average ACC for all frequencies was 83.1%.

For the experiment that used a screen to present the scene, Scenario 3 used a map and the user saw the location of the destination initially, but the ACC was obviously low. Therefore, Scenario 2 was used to confirm the user’s choice and increase the accuracy of the BCI system.

Similar to Scenario 1 using the CCA classifier, the results from using the MSI analysis tool were 94.5%, 85%, 89.6%, 88.8% and 72.9%, respectively. The average ACC for all frequencies was 86.2%. For Scenario 2, using the same frequency and MSI for analysis, the classification ACCs were 94.2%, 91.3%, 90.8%, 79.6% and 70%, respectively. The average ACC for all frequencies was 86.3%. Then the classification rates in Scenario 3 were 92.1%, 84.6%, 82.1%, 77.5% and 68.3%, respectively. The average ACC for all frequencies was 80.9%.

The results for the experiment that used MR goggles and CCA for analysis, collected from the same twelve participants, are shown in Table 7, Table 8 and Table 9 respectively. The experimental results using MSI as the analysis method are shown in Table 10, Table 11 and Table 12. Scenarios 1, 2 and 3 were the same as those for the experiment using the screen. Using MR goggles, Scenario 1, which controlled the direction of the electric wheelchair, had respective accuracies (ACCs) for the four orientations, backward, forward, left and right, and an end option at frequencies of 7 Hz, 8 Hz, 9 Hz, 11 Hz and 13 Hz were 95.8%, 97.9%, 100%, 98.8% and 97.5%. The average ACC for all frequencies was 98%. Scenario 2 used the same frequencies and the classification ACCs were 94.6%, 96.3%, 98.8%, 98.3% and 95.8%, respectively. The average ACC for all frequencies was 96.8%. Scenario 3 used a map of the entire experimental field and the ACCs were 99.6%, 98.8%, 100%, 98.3% and 97.5%, respectively. The average ACC for all frequencies was 98.8%.

In the experiment using MR goggles as a display, the results of analyzing the EEG signal with MSI for Scenario 1 were 97.5%, 98.8%, 100%, 97.1% and 95.4%. The average ACC for all frequencies was 97.8%. Scenario 2 used the same frequencies and the classification ACCs were 96.3%, 93.8%, 98.8%, 95% and 92.1%, respectively. The average ACC for all frequencies was 95.2%. Scenario 3 used a map of the entire experimental field and the ACCs were 99.2%, 98.8%, 98.8%, 96.7% and 88.3%, respectively. The average ACC for all frequencies was 95%.

During an online CCA experiment, each category of the different frequencies generates a confidence score. The CCA classifier generates classification results using these confidence scores so the proposed recognition algorithm determines the highest score to classify this EEG signal to the corresponding frequency. However, if the user does not pay attention to the flickers on the screen or wants to change the mode, the confidence scores are low. A threshold based on these twelve subjects is proposed. For Scenario 1, the threshold, TD, is 0.215 and for Scenarios 2 and 3, the thresholds, TR and TM, have a value of 0.22. Table 13 shows the specified thresholds for this study for the BCI system to enter the next mode.

These experimental results show that the CCA classifier accurately classifies the EEG signals into corresponding classes. Using MR goggles to present flickering stimuli is more accurate and convenient than using a screen to present the scenario. A higher classification rate allows the electric wheelchair to move more stably and safely and the BCI system is easier to use.

### 5.3. ITR Experimental Results

After analyzing the accuracy of the classification results from the CCA classifier was determined, the ITR then determined whether the scenarios for this study were suitable for stimulating the brain.

For each experiment, one selection took three seconds: stimulation took two seconds and there was a rest for one second. Five targets were used for each scenario, with frequencies of 7 Hz, 8 Hz, 9 Hz, 11 Hz and 13 Hz. This study used two types of experiments and three scenarios to determine the accuracy. A screen and MR goggles were used to present the scene.

Table 14 shows the average values for the ACC and the ITR for the proposed three scenarios for the first experiment. The respective ITR values for Scenarios 1, 2 and 3 were 33.79, 31.84 and 26.57 bits per minute, which present the degree of data converting to stimulate, as shown in Table 14. Table 15 shows the mean values for ACC and ITR for the three scenarios for the second experiment. The ITR values were 42.81 bits per minute for Scenario 1, 41.07 bits per minute for Scenario 2 and 44.08 bits per minute for Scenario 3. The ITR distribution graphs for these two experiments are shown in Figure 10 and Figure 11.

## 6. Discussion

According to the experimental results, they can be mainly divided into two parts: (1) This paper mainly uses CCA as a tool for analyzing EEG, and compares it with other SSVEP-based studies such as MSI are shown in Table 16. (2) There were changes in the monitors that display the stimuli and comparisons of their advantages and disadvantages.

The experimental results show that CCA has a higher accuracy in both experiment 1 and experiment 2. It means that the BCI system proposed in this article is feasible and accurate, and has a certain degree of reliability in controlling electric wheelchairs. The other results show that MR goggles give a significantly better accuracy than the screen. The MR goggles are mounted on the subject’s head so the stimulus remains in the line of sight, even if the subject moves their head. The MR goggles also wrap around the eyes so the visual area is not easily disturbed. The position of the MR goggles’ host computer coincides with the visual area of the brain wave cap. The host computer is pressed onto the electrodes so the electrode is not easily displaced when the subject moves their head.

However, the subjects of the current experiment are all healthy people, and the experimental field is not in the hospital, which has limitations on the development of the entire system. However, we cooperated with Mackay Memorial Hospital in Taipei, and we will continue to keep in touch and plan further experiments and recruit more contextual subjects in the future.

## 7. Conclusions

This study proposes an electric wheelchair that is controlled using EEG signals that are acquired using an SSVEP-based BCI system. Firstly, three channels that are related to the visual cortex in a 21-channel EEG electrode cap are used to collect the EEG signals from the operator. The subjects focus on the monitor, which displays the three proposed scenarios.

For Scenario 1, the operator directly controls the electric wheelchair’s direction of motion. Hence, four orientations and a termination choice are displayed for this scenario. Scenarios 2 and 3 are designed to involve automatic control. Scenario 2 shows the information for five rooms. Furthermore, Scenario 3 migrates map information with the environment to Scenario 2.

When the EEG signals are collected, this study analyzes the data using a CCA classifier. The ITR is calculated to evaluate the classification results from the CCA to the stimulus. Finally, the processed EEG signals are then translated into commands to the electric wheelchair to complete the tasks.

The average ACCs for the three scene classifications for the first experiment are 90.67%, 88.82% and 83.08%, respectively. For the experiment using MR goggles to present stimuli, the average ACCs for the three scene classifications are 98%, 96.8% and 98.8%, respectively.

These results show that MR goggles are more effective than a screen to present a scene stimulus. A screen is large and occupies most of the space on the wheelchair, but MR goggles allow the subject to observe, require little space and are not easily disturbed by the outside world. The CCA classification rate is also better for MR goggles.

The accurate results show that the proposed system classifies the EEG signal into the correct category. Furthermore, the electric wheelchair is accurately and safely guided and the BCI system for this study allows the user to reach a specified location easily.

## Figures and Tables

**Figure 1 biosensors-12-00772-f001:**
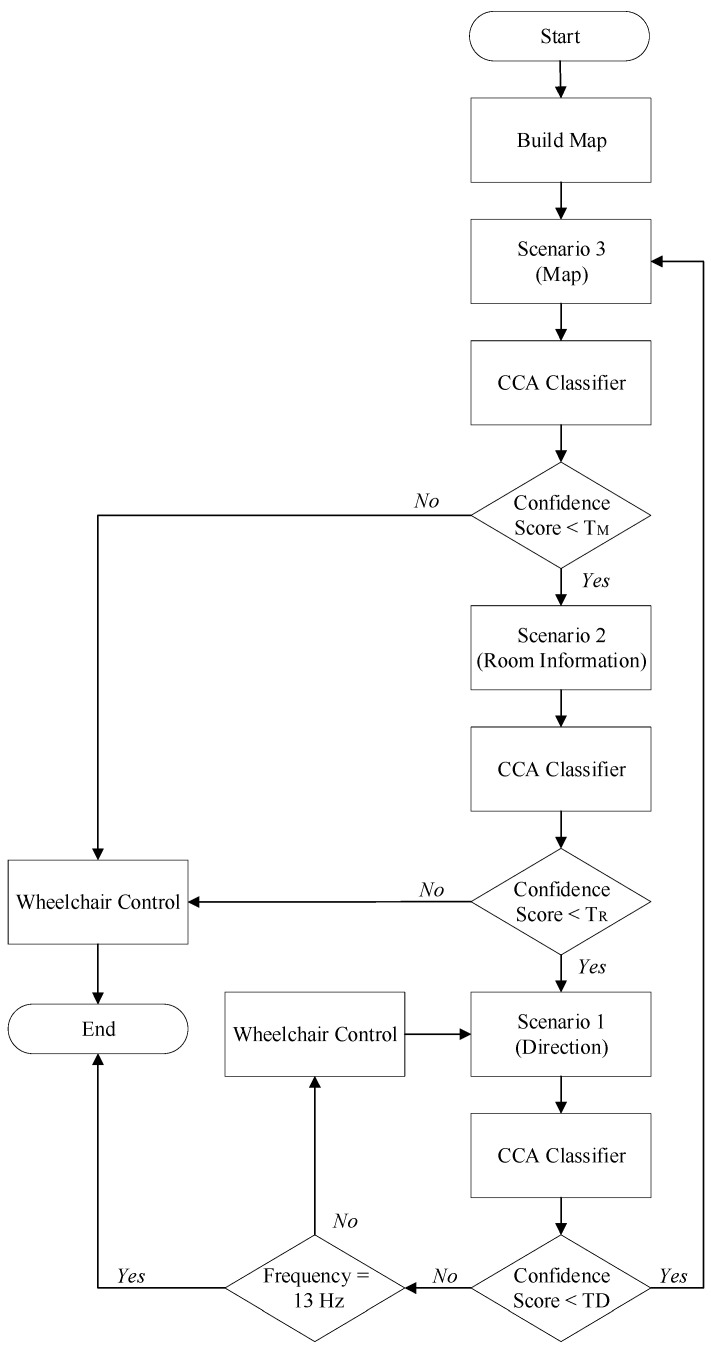
The architecture for this study. Subjects select one of the three scenarios for automatic interactive control. EEG signals are acquired by participants using one of the scenarios. When the data are collected, a CCA classifier detects the target. Analyzed EEG signals are then translated to commands to control the electric wheelchair.

**Figure 2 biosensors-12-00772-f002:**
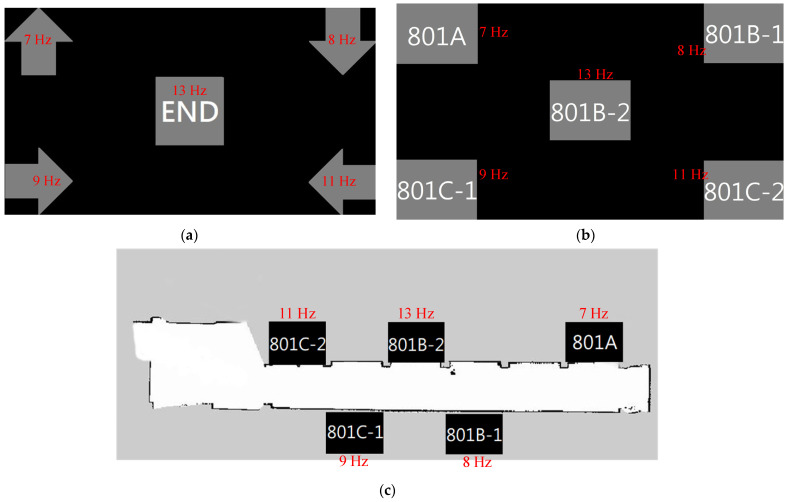
The three scenario configurations for this study: (**a**) Scenario 1 with four directions on the corners and an end choice in the middle of the black background, (**b**) Scenario 2 with five pieces of information of each room and (**c**) Scenario 3, where the map is constructed by the electric wheelchair.

**Figure 3 biosensors-12-00772-f003:**
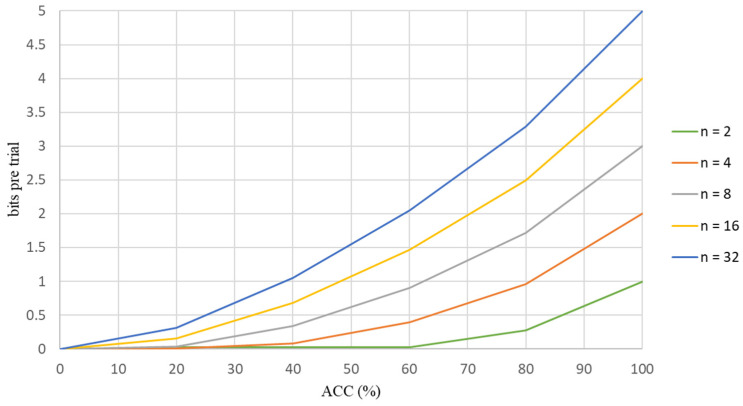
The information that is transferred in bits/trial for different numbers of targets.

**Figure 4 biosensors-12-00772-f004:**
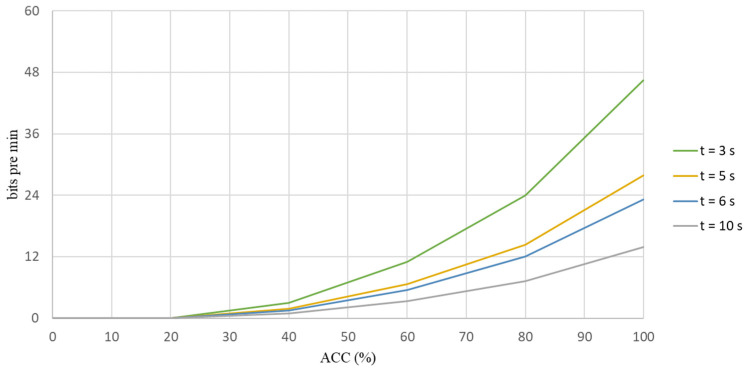
The information that is transferred rate in bits/trial for different average times for one selection.

**Figure 5 biosensors-12-00772-f005:**
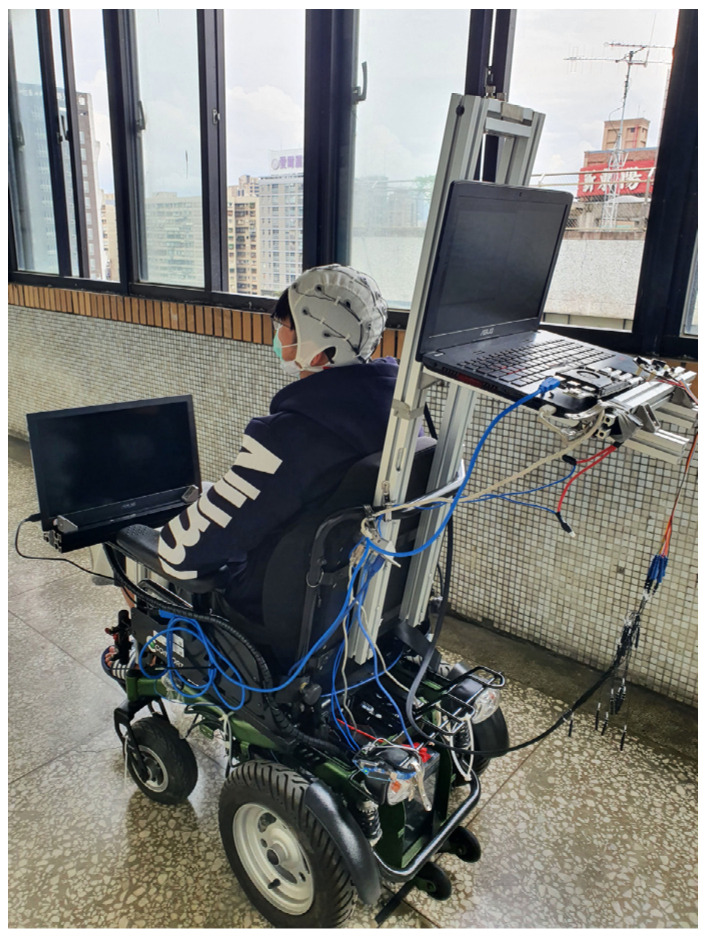
The electric wheelchair for this study.

**Figure 6 biosensors-12-00772-f006:**
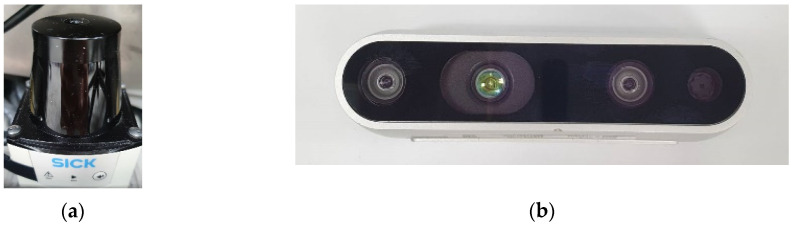
The sensors that are installed on the electric wheelchair: (**a**) SICK TIM551 2D-LiDAR and (**b**) Intel^®^ RealSense D435i RGB-D camera.

**Figure 7 biosensors-12-00772-f007:**
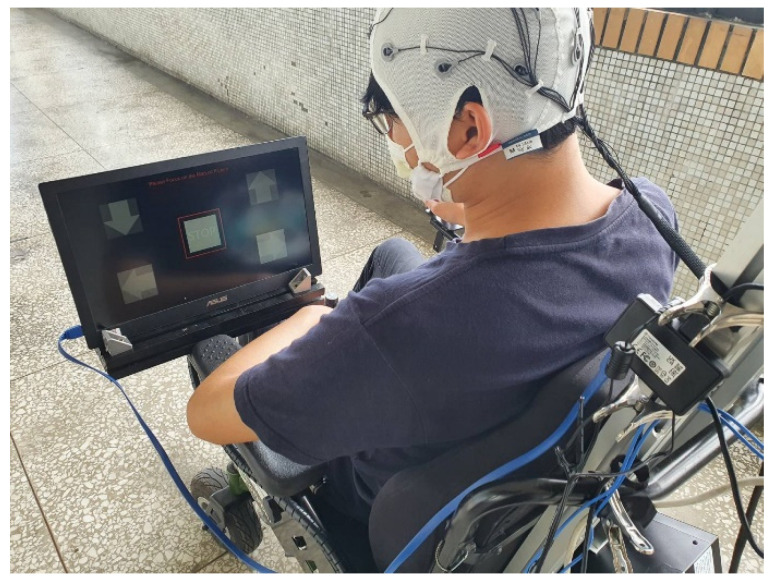
The configuration for the experiment using a screen.

**Figure 8 biosensors-12-00772-f008:**
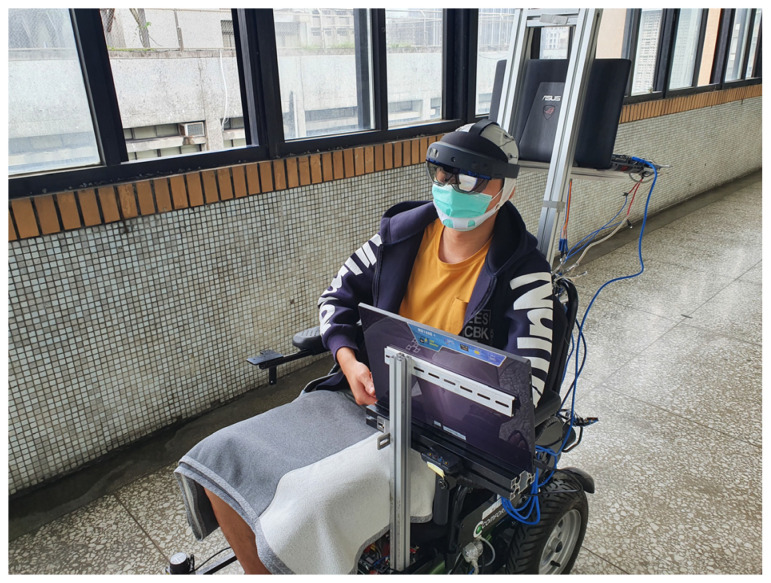
The configuration for the experiment using MR goggles.

**Figure 9 biosensors-12-00772-f009:**
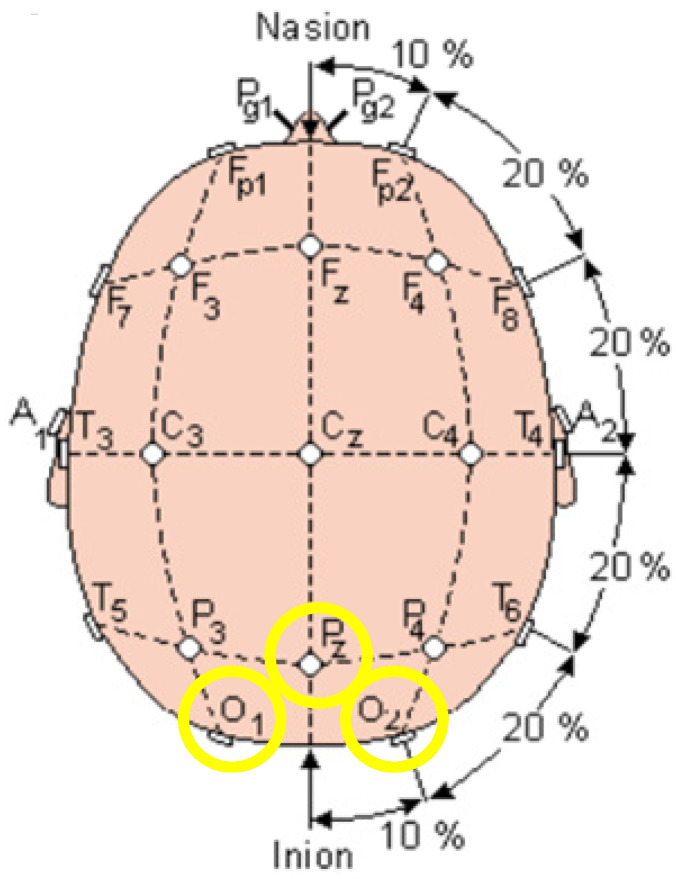
The configuration of channels for the EEG cap [29].

**Figure 10 biosensors-12-00772-f010:**
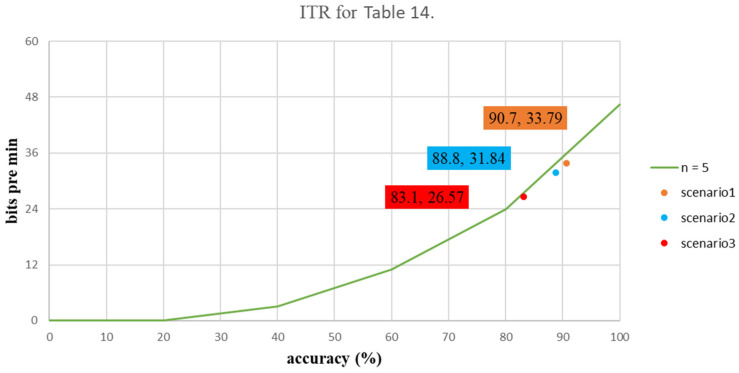
The ITR value for each stimulus using a screen.

**Figure 11 biosensors-12-00772-f011:**
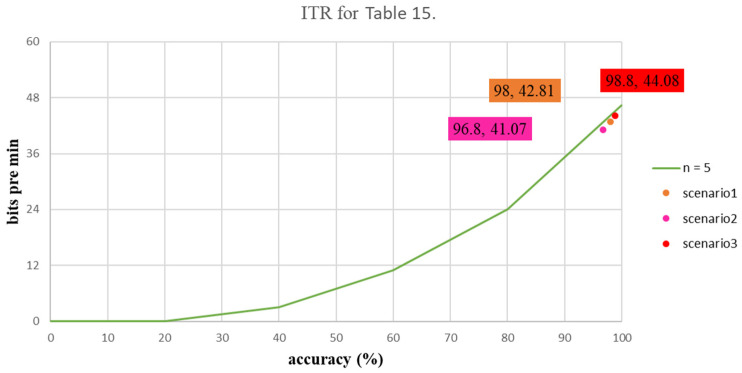
The ITR value for each stimulus using MR goggles.

**Table 1 biosensors-12-00772-t001:** The confusion matrix for the SSVEP recognition results using CCA classifier for the direction test in the first experiment (Scenario 1).

True Class	Predicted Class					ACC (%)
	7 Hz	8 Hz	9 Hz	11 Hz	13 Hz	
7 Hz	228	4	3	1	4	95
8 Hz	12	218	3	4	3	90.8
9 Hz	10	5	217	7	1	90.4
11 Hz	3	4	6	226	1	94.2
13 Hz	13	9	9	10	199	83
Precision (%)	85.7	90.8	91.2	91.1	95.7	

**Table 2 biosensors-12-00772-t002:** The confusion matrix for the SSVEP recognition results using CCA classifier for the room information test in the first experiment (Scenario 2).

True Class	Predicted Class					ACC (%)
	7 Hz	8 Hz	9 Hz	11 Hz	13 Hz	
7 Hz	227	3	7	2	1	94.6
8 Hz	8	221	6	3	3	92.1
9 Hz	3	7	221	6	3	92.1
11 Hz	11	8	6	212	3	88.3
13 Hz	20	12	11	12	185	77.1
Precision (%)	89.8	98	97	98	100	

**Table 3 biosensors-12-00772-t003:** The confusion matrix for the SSVEP recognition results using CCA classifier for the map test in the first experiment (Scenario 3).

True Class	Predicted Class					ACC (%)
	7 Hz	8 Hz	9 Hz	11 Hz	13 Hz	
7 Hz	209	12	6	11	2	87.1
8 Hz	22	198	4	10	6	82.5
9 Hz	13	9	209	6	3	87.1
11 Hz	15	19	6	197	3	82.1
13 Hz	17	16	11	12	184	76.7
Precision (%)	75.7	78	88.6	83.5	92.9	

**Table 4 biosensors-12-00772-t004:** The confusion matrix for the SSVEP recognition results using MSI classifier for the direction test in the first experiment (Scenario 1).

True Class	Predicted Class					ACC (%)
	7 Hz	8 Hz	9 Hz	11 Hz	13 Hz	
7 Hz	227	6	3	1	3	94.5
8 Hz	22	204	9	4	4	85
9 Hz	11	9	215	5	1	89.6
11 Hz	12	7	7	213	2	88.8
13 Hz	27	17	17	8	175	72.9
Precision (%)	75.9	84	85.7	92.2	94.6	

**Table 5 biosensors-12-00772-t005:** The confusion matrix for the SSVEP recognition results using MSI classifier for the room information test in the first experiment (Scenario 2).

True Class	Predicted Class					ACC (%)
	7 Hz	8 Hz	9 Hz	11 Hz	13 Hz	
7 Hz	226	2	5	5	2	94.2
8 Hz	8	219	7	3	3	91.3
9 Hz	7	9	218	4	2	90.8
11 Hz	22	20	6	191	1	79.6
13 Hz	31	17	14	10	168	70
Precision (%)	76.9	84.9	87.2	89.7	95.5	

**Table 6 biosensors-12-00772-t006:** The confusion matrix for the SSVEP recognition results using MSI classifier for the map test in the first experiment (Scenario 3).

True Class	Predicted Class					ACC (%)
	7 Hz	8 Hz	9 Hz	11 Hz	13 Hz	
7 Hz	221	6	5	5	3	92.1
8 Hz	21	203	6	6	4	84.6
9 Hz	23	11	197	7	2	82.1
11 Hz	28	17	5	186	4	77.5
13 Hz	30	20	13	13	164	68.3
Precision (%)	68.4	79	87.2	85.7	92.7	

**Table 7 biosensors-12-00772-t007:** The confusion matrix for the SSVEP recognition results using CCA classifier for the direction test in the second experiment (Scenario 1).

True Class	Predicted Class					ACC (%)
	7 Hz	8 Hz	9 Hz	11 Hz	13 Hz	
7 Hz	230	1	2	4	3	95.8
8 Hz	4	235	1	0	0	97.9
9 Hz	0	0	240	0	0	100
11 Hz	1	1	1	237	0	98.8
13 Hz	1	0	3	2	234	97.5
Precision (%)	97.5	99.2	97.2	91.13	95.67	

**Table 8 biosensors-12-00772-t008:** The confusion matrix for the SSVEP recognition results using CCA classifier for the room information test in the second experiment (Scenario 2).

True Class	Predicted Class					ACC (%)
	7 Hz	8 Hz	9 Hz	11 Hz	13 Hz	
7 Hz	227	2	2	5	4	94.6
8 Hz	3	231	2	3	1	96.3
9 Hz	3	0	237	0	0	98.8
11 Hz	1	1	1	236	1	98.3
13 Hz	2	2	2	4	230	95.8
Precision (%)	96.2	97.9	97.1	95.2	97.5	

**Table 9 biosensors-12-00772-t009:** The confusion matrix for the SSVEP recognition results using CCA classifier for the map test in the second experiment (Scenario 3).

True Class	Predicted Class					ACC (%)
	7 Hz	8 Hz	9 Hz	11 Hz	13 Hz	
7 Hz	239	0	0	1	0	99.6
8 Hz	2	237	0	1	0	98.8
9 Hz	0	0	240	0	0	100
11 Hz	1	2	1	236	0	98.3
13 Hz	1	0	4	1	234	97.5
Precision (%)	98.4	99.2	98	98.7	100	

**Table 10 biosensors-12-00772-t010:** The confusion matrix for the SSVEP recognition results using MSI classifier for the direction test in the second experiment (Scenario 1).

True Class	Predicted Class					ACC (%)
	7 Hz	8 Hz	9 Hz	11 Hz	13 Hz	
7 Hz	234	2	1	2	1	97.5
8 Hz	3	237	0	0	0	98.8
9 Hz	0	0	240	0	0	100
11 Hz	2	2	3	233	0	97.1
13 Hz	4	3	2	2	229	95.4
Precision (%)	96.3	97.1	97.6	98.3	99.6	

**Table 11 biosensors-12-00772-t011:** The confusion matrix for the SSVEP recognition results using MSI classifier for the room information test in the second experiment (Scenario 2).

True Class	Predicted Class					ACC (%)
	7 Hz	8 Hz	9 Hz	11 Hz	13 Hz	
7 Hz	231	3	2	2	2	96.3
8 Hz	7	225	4	2	2	93.8
9 Hz	3	0	237	0	0	98.8
11 Hz	6	2	3	228	1	95
13 Hz	4	7	5	3	221	92.1
Precision (%)	92	94.9	94.4	97	97.8	

**Table 12 biosensors-12-00772-t012:** The confusion matrix for the SSVEP recognition results using MSI classifier for the map test in the second experiment (Scenario 3).

True Class	Predicted Class					ACC (%)
	7 Hz	8 Hz	9 Hz	11 Hz	13 Hz	
7 Hz	238	0	0	1	1	99.2
8 Hz	1	237	0	2	0	98.8
9 Hz	2	1	237	0	0	98.8
11 Hz	5	2	1	232	0	96.7
13 Hz	9	8	8	3	212	88.3
Precision (%)	93.3	95.6	96.3	97.5	99.5	

**Table 13 biosensors-12-00772-t013:** The thresholds that we designed for the BCI system.

	TD	TR	TM
Confidence Score	0.215	0.22	0.22

**Table 14 biosensors-12-00772-t014:** The average accuracy for the direction test (Scenario 1), the room information test (Scenario 2) and the map test (Scenario 3) using a screen with CCA classifier.

Scenario	Average ACC (%)	ITR (bits/min)
1	90.7	33.79
2	88.8	31.84
3	83.1	26.57

**Table 15 biosensors-12-00772-t015:** The average accuracy for the three scenarios using MR goggles with CCA classifier.

Scenario	Average ACC (%)	ITR (bits/min)
1	98	42.81
2	96.8	41.07
3	98.8	44.08

**Table 16 biosensors-12-00772-t016:** Comparison of all experiments.

Scenario	Classifier/Monitor			
	CCA/Screen	CCA/MR Goggles	MSI/Screen	MSI/MR Goggles
1	90.7%	98%	86.2%	97.8%
2	88.8%	96.8%	86.3%	95.2%
3	83.1%	98.8%	80.9%	95%

## Data Availability

Not applicable.

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
