# Peer review of "An Electric Wheelchair Manipulating System Using SSVEP-Based BCI System"

_biosensors, 2022, doi:10.3390/bios12100772_

Round 1

Reviewer 1 Report

The manuscript presents a SSVEP-based BCI system for wheelchair control. Although the topic is interesting and definitely scientifically sound, the manuscript has major flaws. Below are some comments to the authors:

1.      The introduction section needs to be revised. The authors should clearly distinguish BCI systems into invasive/non-invasive and endogenous/ exogenous. Please use as a guidance the following recently published thorough review:

Palumbo, A.; Gramigna, V.; Calabrese, B.; Ielpo, N. Motor-Imagery EEG-Based BCIs in Wheelchair Movement and Control: A Systematic Literature Review. Sensors 202121, 6285. https://doi.org/10.3390/s21186285

2.      The structure of the sections is not well. In the Introduction the authors should briefly describe their proposed work in one (and commonly in the last) paragraph.

3.      I had to reach the Conclusion to find that the electrode cap consists of 21 electrodes. Please be careful on this. I suggest the authors to include this information in the experimental procedure.

4.      The Discussion section is missing. The authors should include a Discussion section describing their main findings, their limitations and also include a table of other similar works and compare their finding with other SSVEP-based studies.

5.      The experiments were performed on 12 subjects. The number is generally limited; however, most of these studies does not utilize data from an appropriate number of subjects. This should be discussed in the discussion section. Most important, are the participants healthy or with physical disability?

Minor

6. Line 66. The CCA abbreviation is used for the first time in the manuscript (apart from Abstract). Use the entire meaning and then the abbreviated form.

Reviewer 2 Report

This work presented an electroencephalography (EEG)-based brain-computer interface (BCI) system and a steady-state visual evoked potential (SSVEP) to manipulate an electric wheelchair. Canonical correlation analysis (CCA) is used to classify the EEG signals into the corresponding class of command. Though the studied task is interesting, some critical comments need to be addressed to improve the manuscript.

1. What type of cross-validation was implemented to evaluate the experimental performance? I would suggest using a K-fold (e.g., 10-fold) cross-validation.

2. The authors should give the readers some concrete information to get them excited about their work. The current abstract only describes the general purposes of the article. It should also include the article's main (1) impact and (2) significance on the relevant field.

3. There is a lack of interpretation of SSVEP patterns identified by the CCA method.

4. Many relevant decoding methods have been recently developed for SSVEP analysis in BCIs. The authors need to review more of those work, such as, Filter bank-driven multivariate synchronization index for training-free SSVEP BCI; Correlated component analysis for enhancing the performance of SSVEP-based brain-computer interface.

5. What are the potential limitations of the method? Please give a discussion on this and also provide several future study directions for addressing these limitations.

Round 2

Reviewer 2 Report

The authors have clarified the reviewers' concerns. I have no further issues to disclose.